**Subject Category:**
Biology (whole organism)

behaviour/evolution/ecology

mice, Trivers–Willard, sex ratio, offspring

**Author for correspondence:**
A. M. Edwards
e-mail: a.edwards@latrobe.edu.au

# Maternal effects obscure condition-dependent sex allocation in changing environments

A. M. Edwards[1,2], E. Z. Cameron[2,3], E. Wapstra[1] and J. McEvoy[1]

[1]Department of Ecology, Environment and Evolution, La Trobe University, Bundoora, Victoria 3086, Australia
[2]School of Biological Sciences, University of Tasmania, Private Bag 55, Hobart, Tasmania 7001, Australia
[3]School of Biological Sciences, University of Canterbury, Private Bag 4800, Christchurch, New Zealand

AME, 0000-0003-0005-4650

Climate change increases environmental fluctuations which thereby impact population demography. Species with temperature-dependent sex determination may experience more extreme sex ratio skews, but this has not been considered in species with chromosomally determined sex. However, anticipatory maternal effects cause lifelong physiological changes impacting sex ratios. Here we show, in mice, that more sons were born to mothers in good condition when their breeding environment matched their gestational environment, consistent with theoretical predictions, but mothers in mismatched environments have no condition–sex ratio relationship. Thus, the predicted effect of condition on sex ratio was obscured by maternal effects when the environment changed. This may explain extreme sex ratio skews in reintroduced or translocated populations, and sex ratio skews may become more common and less predictable with accelerating environmental change.

## 1. Introduction

Maternal effects are defined as the causal influences of the mother's phenotype or genotype on developing offspring [1] and can have profound effects on offspring life history through, for example, lifelong physiological changes in offspring [2–4]. During gestation, the mammalian mother in particular has a prolonged period of contact during which the environment that the mother experiences can interact with the development of the offspring, particularly through the uterine environment, thereby affecting the offspring's

phenotype [3]. These maternal effects may be developmental, or the previous experiences of the parents may also be transmitted epigenetically [5]. Furthermore, the uterine environment can be influenced by the offspring's siblings, similarly causing physiological changes [6]. Therefore, the uterine environment can have extensive, long-lasting influences on the offspring through both maternal and sibling effects.

Pre-programming of offspring to environmental conditions through maternal effects can be advantageous, as it allows phenotypic plasticity of offspring to occur at a faster rate than would be seen by adaptation through natural selection [7]. For example, snowhoe hares (*Lepus americanus*) appear to show prenatal glucocorticoid programming which influences their baseline stress levels and susceptibility to stressors, preparing them for the stress levels experienced by their mothers [8]. Conversely, those born in low predation years exhibited higher stress levels [8]. However, environments are not static and therefore, the environment that the mother experiences during gestation may not be the same as the post-natal environment that the offspring experiences, which may result in decreased offspring fitness (reviewed in [9]). For example, high predation from lynx is linked to crashes in the snowshoe hare populations which then remain low, despite the removal of the threat, due to intergenerational, maternally inherited stress hormones from the population decline period [8]. Therefore, the mismatch between pre- and post-natal environments can be detrimental to offspring [10]. Artificially simulated increases or decreases in maternal stress during late gestation can result in a mismatched stress response in offspring, which may lead to abnormal predator responses [10], increased anxiety behaviours [11] and decreased cognitive abilities [12]. Such alterations may then impact other life-history traits, including survival and reproductive success (e.g. [8]).

Stress physiology has been linked mechanistically to sex allocation, both directly and through an interaction with glucose (e.g. [13–15]). Stress causes a male bias in litters [16,17], which has been experimentally reversed using dexamethasone, a synthetic glucocorticoid [18]. The reversal of the litter bias was attributed to a reduction in stress caused by dexamethasone, but since stress results in higher levels of circulating glucose [19], it is possible that changes in glucose are more directly responsible for the sex ratio changes. Male and female conceptus are sexually dimorphic in their susceptibility to glucose levels [20], with increased glucose levels favouring male conceptus growth and development [21]. It has been shown mechanistically that application of dexamethasone at conception results in a decrease in circulating glucose, and as expected a female bias in the resultant litter [13].

Hypotheses of sex ratio adjustment predict that parents should adjust the sex ratio of their offspring with local conditions or ability to invest, if net fitness returns are sex-specific [22–25]. For example, directional sex allocation is predicted where one sex is differentially advantaged in reproductive success by extra investment [25]. Generally, studies support the trend [26,27], but there remains a level of unexplained variation [28], and unpredictable effect sizes between individuals [29]. This variation suggests the possibility of constraints imposed on a female's ability to respond to the environment [30]. Changes to baseline physiology as a result of maternal effects may explain some of the inter-individual variation [31].

Recently, we conducted a study on laboratory mice that used oral application of dexamethasone, to experimentally induce an altered-stress gestational environment. Dexamethasone, when applied to a mother during late gestation, caused physiological changes in the stress response of her female offspring. These physiological changes decreased the female's offspring sex ratio resulting in more daughters under normal environmental conditions, due to a mismatch between her pre- and post-natal environments [31]. The same experimentally induced low-stress environment experienced only at the time of conception also decreased her offspring sex ratio [13]. Given that treatment during gestation and at conception results in a female-biased sex ratio in subsequent litters, it could be hypothesized that application of the treatment at both time points would have an additive effect. However, considering the sex ratio bias is likely due to a mismatch in maternal effects, the application of both treatments may result in the bias disappearing, as the environments match. Here, we test the effects of these combined prenatal and conception treatments of dexamethasone on laboratory mice. We propose two hypotheses: (1) that the combined treatments result in an additive response of decreased offspring sex ratios, predicted if females are responding independently to each of the environmental treatments, or (2) that the combined treatment results in a negated effect, predicted if the response is due to maternal effects and the pre- and post-natal environments matching.

## 2. Material and methods

We used BALB/c mice bred and housed at the University of Tasmania, Australia. They were kept under 12 L : 12 D photoperiod in a temperature and humidity controlled room and provided with mouse chow

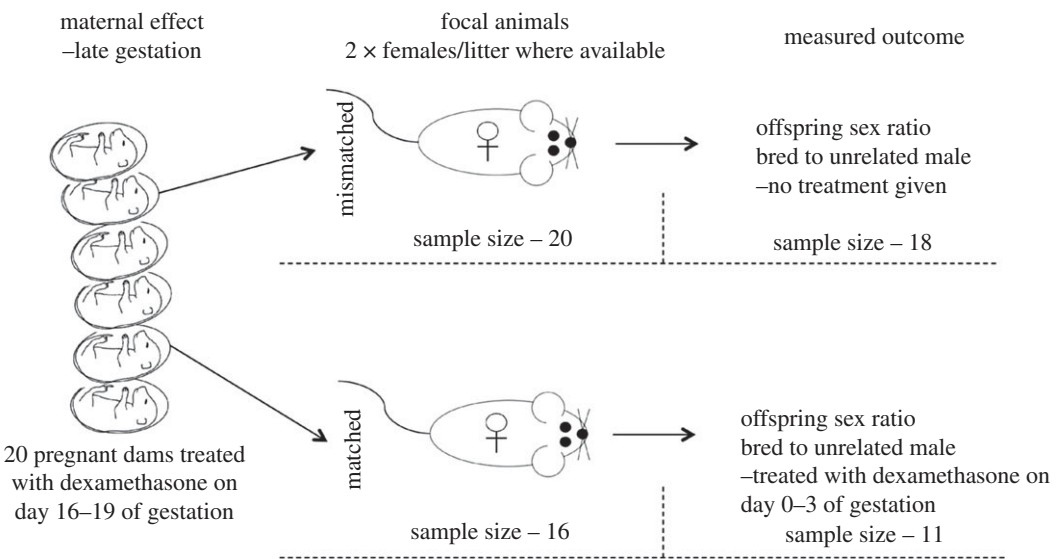

**Figure 1.** The experimental design of a sex allocation study investigating whether maternal effects influence a female's ability to respond to environmental pressure. The sample sizes at each stage of the experiment are listed.

**Table 1.** The list of variables from laboratory mice collected to determine the influence on sex allocation in females with matching and mismatching pre- and post-natal environments. Body measurements were taken at seven weeks old.

| variable | description |
| --- | --- |
| anogenital distance | the distance between the anus and the genital opening, measured using digital callipers. An indicator of prenatal androgen exposure [32] |
| blood glucose | blood was collected via tail tipping, glucose levels were measured using an Accu-Chek Performa Nano glucometer |
| body condition | calculated from the residuals of an ordinary least-squares linear regression of body mass and pes length [33]. Pes length is measured from the base to the tip of the footpad, using digital callipers |
| digit ratio | digit ratio was calculated as the ratio of second to fourth digit on the hind right foot. Digit length is measured using digital callipers from the tip of the toe to the start of the footpad. Observers were blind to the treatment of the animal. A biomarker for prenatal sex steroid exposure [34] |
| sibling sex ratio | the sex ratio of the litter from which the focal female was taken |

(Barastoc® irradiated food) and filtered water *ad libitum*. The control females used in this study are stock female mice from the colony; they have undergone no treatments. The animals used in this experiment were siblings of those in Edwards *et al.* [30]. Table 1 outlines a list of collected variables.

## 2.1. Generating focal females

The experimental design is outlined in figure 1. Twenty nulliparous dams were housed in groups of up to five until seven weeks of age when they were separated into pairs. One male was introduced to each cage and remained with the females until mating was confirmed via the presence of a copulatory plug.

Following the methods outlined in Edwards *et al.* [31], we used dexamethasone to reduce stress in pregnant dams in late gestation. Dexamethasone is a synthetic glucocorticoid that simulated an artificial low-stress environment in the mothers [13,18]. Fetuses are very sensitive to glucocorticoids [35,36], and therefore protective enzymes (e.g. 11 beta-hydroxysteroid dehydrogenase type 2) exists in the placenta to metabolize approximately 80% of naturally occurring glucocorticoids. Dexamethasone, however, is not metabolized by the placenta, and so the effects are expected to be exaggerated [37].

The interaction of dexamethasone with the mother's body and free dexamethasone interacting with the offspring results in a perceived low-stress environment for the offspring.

At day 16 after the confirmation of a copulatory plug, $1.0\ \mu g\ ml^{-1}$ of dexamethasone (as used by [13]) was added to the drinking water of the dams, and this was replaced with fresh water after 3 days. Although this method results in variable dosages, it eliminates any increase in natural GCs from the stress of handling and injections [18], which could potentially negate the treatment [13]. The females were then left to litter without interruption. Two focal females from each litter were kept for the purpose of this study; however, four dams only produced one female. The mismatched treatment was prioritized and therefore this study consisted of 20 mismatched focal females and 16 matched focal females (figure 1).

## 2.2. Breeding of environmentally mismatched focal females

Mismatched females were mated to unrelated males and allowed to birth naturally with pups being sexed by anogenital distance. One female did not conceive, and another committed infanticide prior to offspring sexing and so was removed from the analysis. The final sample size of environmentally mismatched females was 18 (figure 1).

## 2.3. Breeding of environmentally matched focal females

On the day that the environmentally matched female was added to the male's cage for mating, the water was treated with $1.0\ \mu g\ ml^{-1}$ of dexamethasone, which remained in the cage with the female until day 3 after the presence of a copulatory plug was noted. This treatment simulated a low-stress environment and therefore matched that of the prenatal environment. The females were then allowed to birth naturally and pups were again sexed using anogenital distance. Two females did not conceive, and three others committed infanticide prior to offspring sexing and so were removed from the analysis. The final sample size of environmentally matched females was 11 (figure 1).

## 2.4. Statistics

We used generalized linear models (GLM) with binomial error and an intercept of 1 to verify whether the sex ratios of the two treatment groups and control laboratory mice differed from parity. Results presented are the 95% confidence intervals on the estimate. Significant results are depicted by those intervals that do not include zero.

We used a multivariate analysis of variance (MANOVA) to determine whether the treatment had an effect on any physical body measurement. We used an analysis of variance (ANOVA) to determine whether the litter size between the matched and mismatched mice varied. We also used a generalized linear mixed model with binomial error to investigate the effects of environmental matching and body condition on offspring sex ratio, while accounting for dam ID. The original model included treatment, blood glucose, sibling sex ratio, anogenital distance, body condition and digit ratio as fixed effects and dam ID as the random effect. Using stepwise model simplification, the most parsimonious model included only treatment and body condition, and their interactive effect, along with dam ID. Further to this, the data were then divided into subsets for each of the treatments, and GLM were run to investigate the effect of body condition on each of the treatment groups' offspring sex ratios.

All analyses were performed in R version 3.2.2 [38].

# 3. Results

The environmentally mismatched mice had sex ratios that were significantly lower than the expected 50 : 50 ratio (GLM: $-0.839$, $-0.116$; figure 2); however, neither the environmentally matched group (GLM: $-0.492$, $0.492$) nor the control mice (GLM: $-0.657$, $0.239$) differed from parity. The treatment did not influence physical body measurements ($F_{1,27} = 24.0$, $Pr(>F) = 0.45$), or litter size ($F_{1,27} = 2.46$, $Pr\ (>F) = 0.13$).

When we incorporated condition, there was no effect individually of either treatment ($Z_{1,27} = 1.44$, $P(<Z) = 0.15$) or body condition ($Z_{1,26} = 0.03$, $P(>Z) = 0.97$). However, there was a significant interaction between the two terms ($Z_{1,25} = 2.00$, $P(>Z) = 0.045$; figure 3). Body condition influenced

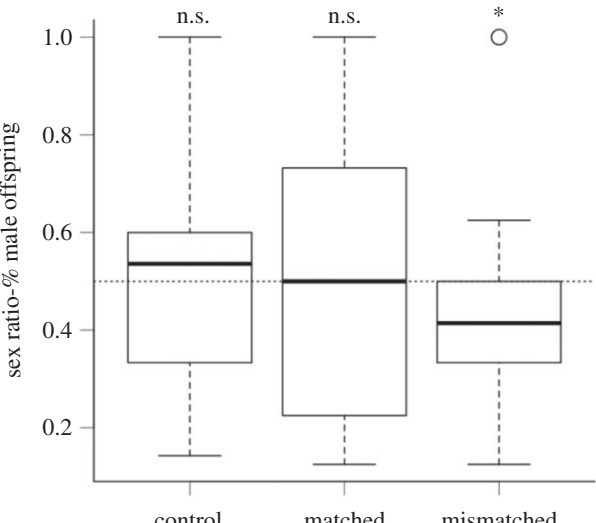

**Figure 2.** The offspring sex ratios from control laboratory mice and those mice whose pre- and post-natal environments match are not significantly different from parity. While those mice whose pre- and post-natal environments do not match have sex ratios that are significantly lower than parity. Note that '*' signifies a significant difference and 'n.s.' signifies a non-significant difference from the expected 50 : 50 ratio. The dotted line indicates the expected 50 : 50 ratio.

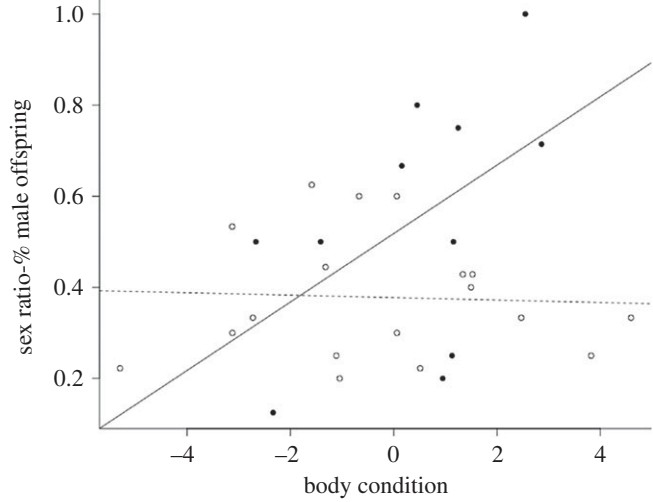

**Figure 3.** The offspring sex ratios from mice whose pre- and post-natal environments match shows a strong positive relationship with body condition, while those with mismatching pre- and post-natal environments do not show any relationship with body condition. Full circles and solid line depict matching environments, open circles and dashed line depict mismatching environments.

sex ratio in the matched group with more sons born to mothers in better condition ($Z_{1,10} = 2.21$, $P(<Z) = 0.03$), but not in the mismatched group ($Z_{1,17} = -0.03$, $P(<Z) = 0.97$; figure 3).

## 4. Discussion

When female mice were treated with dexamethasone both prenatally, and later at conception, the resultant offspring sex ratio was not different from the expected 50 : 50 ratio. Control laboratory female mice also produce sex ratios not different from the expected 50 : 50 ratio; however, female mice treated only once, either prenatally or later at conception, will produce female-biased litters. This allows the rejection of hypothesis 1, where the sex ratio is a direct result of the treatment, and therefore treating at both time points would have been additive, producing a strong female bias. These results are in line with hypothesis 2, that the offspring sex ratio is a result of maternal effects-driven environments and has resulted in no offspring bias. When treated only prenatally, there is a mismatch between the

prenatal and post-natal environment that results in female mice producing biased offspring sex ratios. Our results suggest that treating at the time of conception creates the same environment for which the animal was 'pre-programmed' for, and therefore as the environments match, there is no need for sex ratio adjustment to occur. The prenatal treatment may have caused lifelong physiological changes in the females who are now incapable of producing equal sex ratios under normal laboratory conditions; these physiological changes are not reflected in morphology, as no physical body changes were detected between the treatment groups.

When the environment changed, there was no relationship between maternal body condition and offspring sex ratio, but when gestational and conception environments matched, the predicted condition-dependent sex allocation in line with the Trivers–Willard hypothesis [25] was observed. This supports our hypothesis that anticipatory maternal effects can result in a constraint on maternal sex allocation when the environment changes.

We previously suggested [31] that lowered luteal cortisol [39] and subsequently lowered levels of gluconeogenesis [40] caused changes in free glucose levels [37] influencing offspring sex ratios [26]. Lowered maternal stress levels during late gestation program the physiology of the offspring to be at its optimum in a matching environment [10]. Therefore, using the same dosage of dexamethasone presented in the same manner at conception time, to lower the female's stress levels, should mirror the same environment that she was programmed for, and therefore, we would expect to see that the sex ratio of offspring remains at parity. Sex allocation theory suggests that parents should adjust sex ratios in relation to current local conditions or ability to invest [22,24,25,41], which females were able to do in a matched environment, but not when the stress environment had changed. A changed environment constrained the female's ability to respond to environmental conditions as predicted by sex allocation theory [30].

The mismatch between the maternal environment and the conception environment in the mismatched mice results in two effects that would have been misinterpreted if we had not known the gestational experience of the mice; (a) a significant female bias overall, and (b) no relationship between condition and sex ratio. Previous studies have assumed that all mothers are similarly able to adjust the sex ratio in line with hypothetical predictions, but our study indicates that physiological constraints show that this assumption is likely unjustified in many cases and help to explain the inconsistent results of field studies of sex allocation. Previous studies have shown that anticipatory maternal effects are advantageous when offspring are born into that same environment but are disadvantageous when the environment changes, resulting in population-level effects (e.g. snowshoe hare recovery after lynx die-off [8]). Mismatched maternal effects may therefore remove the adaptive benefits of maternal programming.

Our finding has implications for the management of a variety of species, particularly with interventional management. Translocated individuals or individuals shifted between captive and wild populations may show unexpected reproductive responses due to the mismatch between their development during gestation and adult reproductive environment, possibly contributing to male-biased sex ratios observed in founder populations after reintroduction [42,43]. Furthermore, with increasingly variable environments due to climate change, subtle effects on reproduction may become more marked and contribute to unexpected breeding outcomes in a variety of populations.

Ethics. All experiments were performed under permits granted from the University of Tasmania Animal Ethics Committee (permit nos. A12366 and A13748).

Data accessibility. Data available from the Dryad Digital Repository at: https://doi.org/10.5061/dryad.56451sc [44].

Authors' Contributions. A.M.E. designed and coordinated the study, maintained the animals, carried out the experimental procedures, completed the statistical analysis and drafted the manuscript. E.Z.C. conceived the study, participated in the design of the study, assisted with statistical analysis and helped draft the manuscript. E.W. and J.M. participated in the design of the study and helped draft the manuscript. All authors gave final approval for publication.

Competing Interests. We have no competing interests.

Funding. This work was supported by the Australian Research Council (DP140103227) to E.Z.C. and E.W.

Acknowledgements. We would like to thank Lauren Richards for assistance with mouse husbandry and Paul Scowen for interesting discussions.

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
