## [Reviewer comments · Royal Society Open Science]

Review History

RSOS-181885.R0 (Original submission)

Review form: Reviewer 1 (Marco Festa-Bianchet)

Is the manuscript scientifically sound in its present form?

Yes

Are the interpretations and conclusions justified by the results?

Yes

Is the language acceptable?

No

Is it clear how to access all supporting data?

Yes

Do you have any ethical concerns with this paper?

No

Have you any concerns about statistical analyses in this paper?

No

Recommendation?

Accept with minor revision (please list in comments)

Comments to the Author(s)

I have reviewed an earlier submission of this ms and most of my concerns were addressed. I remain convinced that this experiment is an interesting contribution to our knowledge of adaptive offspring sex ratio manipulation. I have a few minor comments, mostly about the writing:

L. 65-67: this is a somewhat simplistic representation of the hare-lynx dynamics. All that is said here is correct, but it is not the sole driver of the cycle, therefore the wording should be more cautious.

L. 98-101: Here the wording should be modified to clarify that the female offspring are the focus - the current wording may suggest an effect on the treated mothers.

L. 113-114: #2 is not presented as an hypothesis - what would be the prediction?

L. 119-122: Please stick to one tense. Other sections of the paper also switch from present to past.

L. 220: I do not understand what "has resulted in the bias disappearing".

Marco Festa-Bianchet

Review form: Reviewer 2

Is the manuscript scientifically sound in its present form?

Yes

Are the interpretations and conclusions justified by the results?

Yes

Is the language acceptable?

Yes

Is it clear how to access all supporting data?

Yes

Do you have any ethical concerns with this paper?

No

Have you any concerns about statistical analyses in this paper?

No

Recommendation?

Accept as is

Comments to the Author(s)

Greatly improved and now much clearer - a nice paper with important results. As Ref 2 points out, sample sizes are small - but results are convincing. Can be published as it stands

Decision letter (RSOS-181885.R0)

14-Feb-2019

Dear Professor Edwards

On behalf of the Editors, I am pleased to inform you that your Manuscript RSOS-181885 entitled "Maternal effects obscure condition-dependent sex allocation in changing environments" has been accepted for publication in Royal Society Open Science subject to minor revision in accordance with the referee suggestions. Please find the referees' comments at the end of this email.

The reviewers and handling editors have recommended publication, but also suggest some minor revisions to your manuscript. Therefore, I invite you to respond to the comments and revise your manuscript.

- Ethics statement

- Data accessibility

If you wish to submit your supporting data or code to Dryad (<http://datadryad.org/>), or modify your current submission to dryad, please use the following link:
<http://datadryad.org/submit?journalID=RSOS&manu=RSOS-181885>

- Competing interests

- Authors' contributions

- Acknowledgements

- Funding statement

Because the schedule for publication is very tight, it is a condition of publication that you submit the revised version of your manuscript before 23-Feb-2019. Please note that the revision deadline will expire at 00.00am on this date. If you do not think you will be able to meet this date please let me know immediately.

- 1) A text file of the manuscript (tex, txt, rtf, docx or doc), references, tables (including captions) and figure captions. Do not upload a PDF as your "Main Document";
- 2) A separate electronic file of each figure (EPS or print-quality PDF preferred (either format should be produced directly from original creation package), or original software format);

- 3) Included a 100 word media summary of your paper when requested at submission. Please ensure you have entered correct contact details (email, institution and telephone) in your user account;
- 4) Included the raw data to support the claims made in your paper. You can either include your data as electronic supplementary material or upload to a repository and include the relevant doi within your manuscript. Make sure it is clear in your data accessibility statement how the data can be accessed;
- 5) All supplementary materials accompanying an accepted article will be treated as in their final form. Note that the Royal Society will neither edit nor typeset supplementary material and it will be hosted as provided. Please ensure that the supplementary material includes the paper details where possible (authors, article title, journal name).

on behalf of Dr Ryan Earley (Associate Editor) and Kevin Padian (Subject Editor)
openscience@royalsociety.org

Reviewer comments to Author:
Reviewer: 1

Comments to the Author(s)

I have reviewed an earlier submission of this ms and most of my concerns were addressed. I remain convinced that this experiment is an interesting contribution to our knowledge of

adaptive offspring sex ratio manipulation. I have a few minor comments, mostly about the writing:

L. 65-67: this is a somewhat simplistic representation of the hare-lynx dynamics. All that is said here is correct, but it is not the sole driver of the cycle, therefore the wording should be more cautious.

L. 98-101: Here the wording should be modified to clarify that the female offspring are the focus - the current wording may suggest an effect on the treated mothers.

L. 113-114: #2 is not presented as an hypothesis - what would be the prediction?

L. 119-122: Please stick to one tense. Other sections of the paper also switch from present to past.

L. 220: I do not understand what "has resulted in the bias disappearing".

Marco Festa-Bianchet

Reviewer: 2

Comments to the Author(s)

Greatly improved and now much clearer - a nice paper with important results. As Ref 2 points out, sample sizes are small - but results are convincing. Can be published as it stands

Author's Response to Decision Letter for (RSOS-181885.R0)

See Appendix A.

Decision letter (RSOS-181885.R1)

05-Mar-2019

Dear Professor Edwards,

I am pleased to inform you that your manuscript entitled "Maternal effects obscure condition-dependent sex allocation in changing environments" is now accepted for publication in Royal Society Open Science.

Royal Society Open Science operates under a continuous publication model (<http://bit.ly/cpFAQ>). Your article will be published straight into the next open issue and this will be the final version of the paper. As such, it can be cited immediately by other researchers.

As the issue version of your paper will be the only version to be published I would advise you to check your proofs thoroughly as changes cannot be made once the paper is published.

on behalf of Dr Ryan Earley (Associate Editor) and Kevin Padian (Subject Editor)
openscience@royalsociety.org

Associate Editor Comments to Author (Dr Ryan Earley):
Associate Editor: 1
Comments to the Author:
(There are no comments.)

Reviewer comments to Author:

Appendix A

Royal Society Open Science Editorial Office
22/02/2019

Dear Andrew Dunn,

Thank you for taking the time to consider our manuscript. We have made all of the changes suggested by the reviewer and have also amended the statements as requested by yourself. You should find everything completed accurately in the submission. Please let me know if there is anything outstanding. Responses to reviewer comments below in italics.

Kind Regards,
Amy Edwards

Reviewer comments to Author:
Reviewer: 1

Comments to the Author(s)

I have reviewed an earlier submission of this ms and most of my concerns were addressed. I remain convinced that this experiment is an interesting contribution to our knowledge of adaptive offspring sex ratio manipulation.

We thank Marco for his time and effort in reviewing our comments and have addressed all of them completely.

I have a few minor comments, mostly about the writing:

L. 65-67: this is a somewhat simplistic representation of the hare-lynx dynamics. All that is said here is correct, but it is not the sole driver of the cycle, therefore the wording should be more cautious.

This has been changed to a more cautious explanation. "high predation from lynx is linked to crashes in the snowshoe hare populations"

L. 98-101: Here the wording should be modified to clarify that the female offspring are the focus - the current wording may suggest an effect on the treated mothers.

We have adjusted the sentence to clarify that offspring are the focus. "Recently, we conducted a study on laboratory mice that used oral application of dexamethasone, to experimentally-induce a low stress gestational environment. Dexamethasone when applied to a mother during late gestation caused physiological changes in the stress response of her female offspring"

L. 113-114: #2 is not presented as an hypothesis - what would be the prediction?

We have adjusted and added in the prediction. "We propose two hypotheses, 1) that the combined treatments result in an additive response of decreased offspring sex ratios, predicted if females are responding independently to each of the environmental treatments, or 2) that the combined treatment results in a negated effect, predicted if

the response is due to maternal effects and the pre- and post-natal environments matching.”

L. 119-122: Please stick to one tense. Other sections of the paper also switch from present to past.

We have adjusted all places that we can find to past tense.

L. 220: I do not understand what "has resulted in the bias disappearing".

This has been changed to “These results are in line with hypothesis 2, that the offspring sex ratio is a result of maternal effects driven environments and has resulted in no offspring bias”

Marco Festa-Bianchet

Reviewer: 2

Comments to the Author(s)

Greatly improved and now much clearer - a nice paper with important results. As Ref 2 points out, sample sizes are small - but results are convincing. Can be published as it stands

We thank reviewer 2 for their time and comments.

22 Feb 2019

Dr Amy Edwards
Post-Doctoral Researcher

DATE

La Trobe University
Department of Ecology,
Environment and Evolution
Bundoora, Melbourne
Victoria Australia

T +61 3 9479 2245
M +61 4 1888 0975
a.edwards@latrobe.edu.au